# Internal Structure and Reactivations of a Mass Movement: The Case Study of the Jacotines Landslide (Champagne Vineyards, France)

Nicolas Bollot [1],*, Guillaume Pierre [1], Gilles Grandjean [2], Gilles Fronteau [1], Alain Devos [1] and Olivier Lejeune [1]

1   GEGENAA—EA 3795, Université de Reims Champagne-Ardenne, 57 rue Pierre Taittinger, 51571 CEDEX Reims, France
2   BRGM, Risk Division, 3 Avenue Claude Guillemin, CEDEX 02, 45060 Orléans, France
*   Correspondence: nicolas.bollot@univ-reims.fr

**Abstract:** The Jacotines landslide is representative of the large mass movements that affect the Champagne vineyards. Understanding the subsurface structure of these slopes and the mechanisms leading to sliding events is of a great interest, particularly for winegrowers who produce Champagne. This knowledge is generally used to elaborate accurate hazard assessment maps, which is an important feature in land use planning. The approach presented is based on the integration of geophysical imaging (seismic wave velocity and electrical resistivity), lithostratigraphic analysis (drilling core) and geomorphological investigations (surface landforms) to reconstruct the relations between the landslide structure, surface water flow, groundwater regime and the overall slope stability. A first phase of instability resulting in a large rotational slip probably occurred during the Late Glacial Period in morphoclimatic conditions characterized by an excess of water. A second one, still active, leading to superficial reactivations and relates to present hydrogeological conditions determined by the internal structure of the landslide.

**Keywords:** mass-movement; internal structure; geophysics; groundwater; reactivation; Champagne vineyards

## 1. Introduction

In the Montagne de Reims and the Marne valley area (about 450 km$^2$), more than 260 landslides have been identified despite the low relief [1] (Figure 1). Although no precise dating is available in the area, these landslides are generally attributed either to the Late Glacial period (melting of the permafrost and positive hydrological-balance budget) or to the Atlantic and Subboreal period (excess of water), both favorable climatic periods to trigger mass movements in Europe [2–7]. Resultant ancient and dormant landslides of the Champagne vineyards are regularly affected by partial and superficial reactivations which is a problem for winegrowers. The triggering of these reactivations is often attributed to climatic events [8,9], but the causes are generally complex and result mainly from the groundwater conditions [10].

The internal structure of the landslides in the Champagne vineyard area is not well known, though it represents a major key to understand triggering factors and reactivation modalities [10–13]. Both of these points are essential for the risk management and the sustainability of the winegrowing. In recent studies conducted elsewhere than in Champagne area, this information is obtained thanks to multidisciplinary approaches [14–17]. The aim of this work is to produce an interpretative landslide model in the Champagne vineyards based on the internal structure and the hydrological behavior of the hillslopes. The Jacotines landslide, whose morphology is largely representative of the study area (Figure 1), is taken as a pilot site offering the possibility to test a multidisciplinary approach

integrating geomorphology, geophysics and core drilling analysis. This cross-analysis, makes it possible to understand the spatial distribution of springs on the valley-side slopes of Marne River and the relation between groundwater dynamics and mass movements. Previously unexplored in Champagne vineyards, this approach can be transposed to other lowland regions.

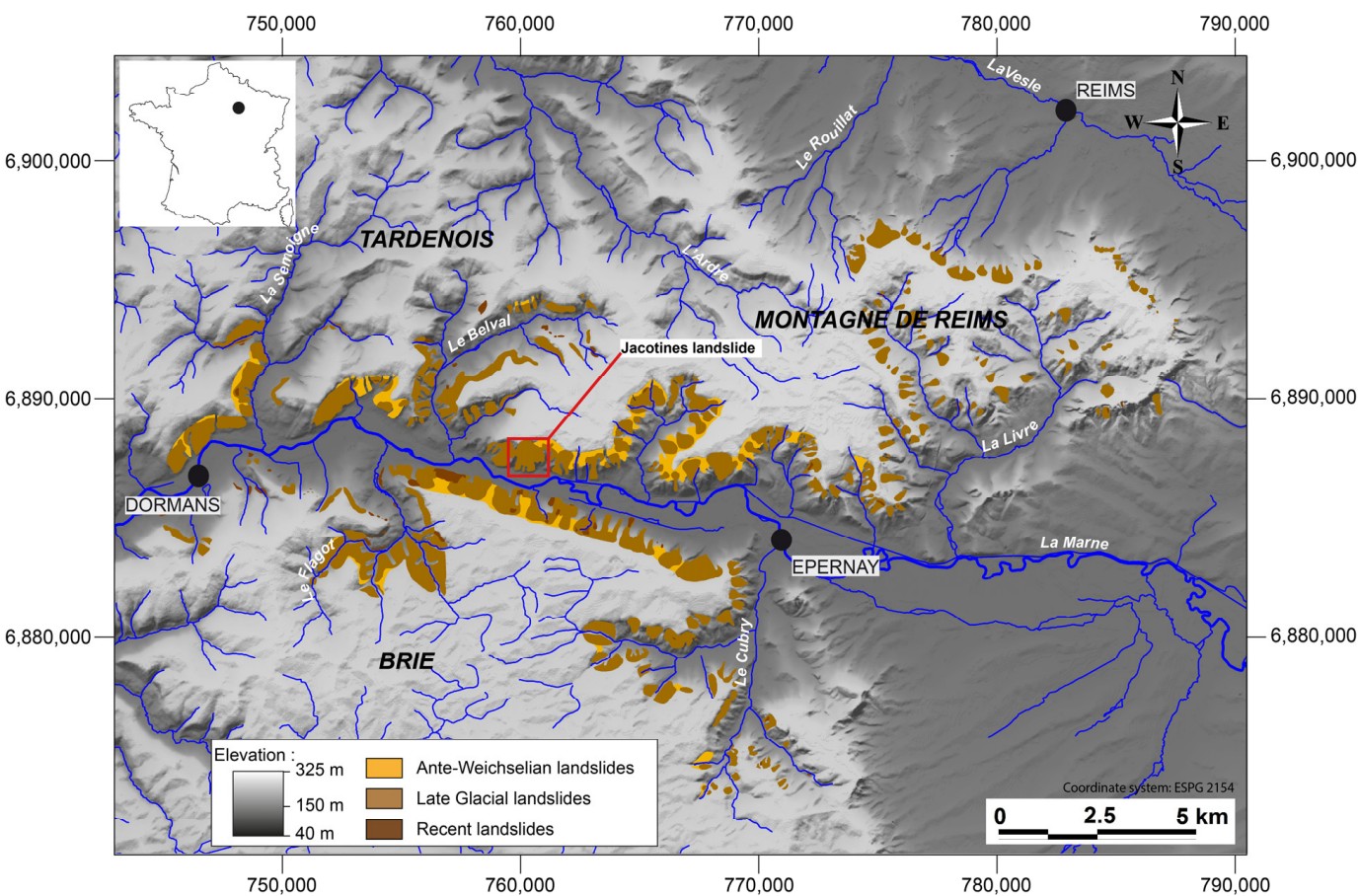

**Figure 1.** Inventory of the landslides of the Marne valley and the Montagne de Reims.

## 2. Study Area and Methods

### 2.1. Study Area

The hillslopes of the region are largely affected by mass movements, some of which of ancient origin (mainly along the Marne valley). However, most of them are of late glacial age and the latest ones occurred during the Holocene (Figure 1). The Jacotines landslide is a typical large rotational landslide. It is located in the commune of Reuil, a few kilometers downstream of the Marne cataclinal breach, between the Montagne de Reims (in the North) and the Brie plateau (in the South; Figure 1).

The base of the slope is cut in Campanian chalk (Upper Cretaceous) which crops out up to 30 m above the bottom of the valley floor. The chalk is overlain by various layers of aclinal cenozoïc rocks: Thanetian sands, Lower Ypresian clays and marls, Upper Ypresian sands, Lutetian limestones and marls, Bartonian and Priabonian clays [18]. The top of the plateau is covered with siliceous limestone and clay derived from Oligocene silicification and from post-exhumation Quaternary argillization (Figure 2).

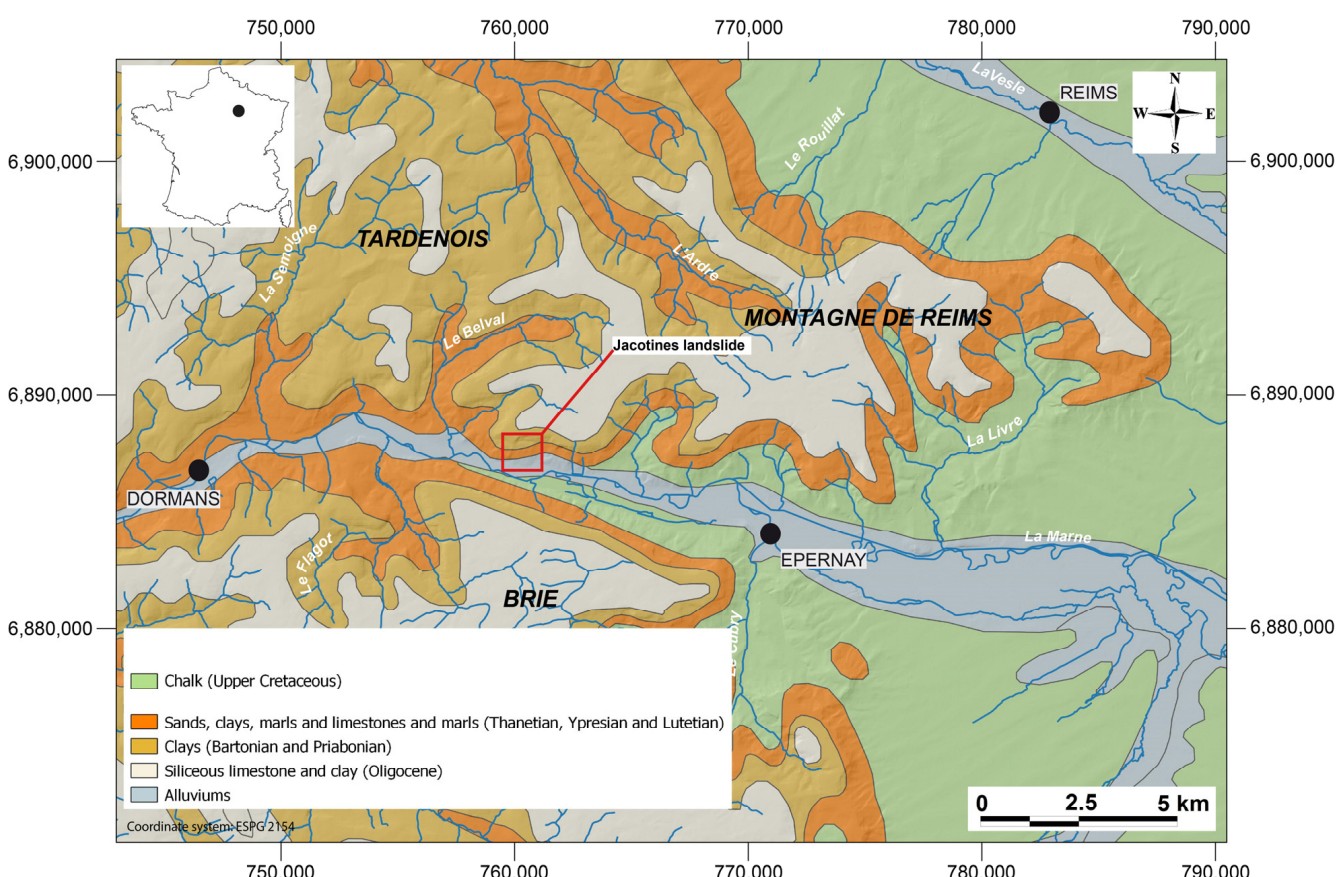

**Figure 2.** Geological map of the Marne valley and the Montagne de Reims.

### 2.2. Material and Methods

First, following Guérémy and Marre [19] and Moeyersons et al. [20], a large scale geomorphological map of the Jacotines landslide (1/10,000) was produced to delimit surface morphology and to localize springs. The internal structure of the landslide was then investigated using geophysical methods. On the one hand seismic refraction tomography data (profiles 1, 2 and 3 on Figure 3) can provide structural information [21–24]. On the other hand electrical resistivity tomography data (profile 3, dipole-dipole and Wenner-Schlumberger methods) can provide information on water content [25–34]. The two methods are also complementary since the seismic data are more reliable in depth while the electrical data give more accurate results near the surface [23]. Measurements were made on the Jacotines landslide along three profiles: two transverse to hillslope direction (profiles 1 and 3, 950 m long and 350 m long respectively) and one longitudinal to hillslope direction (profile 2, 250 m long; Figure 3). Seismic investigations were performed along the three profiles and electrical investigations were performed on profile 1. The pattern of the vineyard plots together with the narrowness of the agricultural paths prevented the realization of a complete longitudinal profile from top to bottom of the landslide.

Geophones and electrodes were set up along the profiles at 5 m interval, and shots were fired at 10 m interval (seismic surveying). These measurements provided information on the distribution of the seismic wave velocity (Vp) and electrical resistivity (ρ). The raw data of profile 1, the most complete, were inverted to obtain 2D tomograms [35,36] by using first breaks traveltimes and apparent electrical resitivities inversion codes.

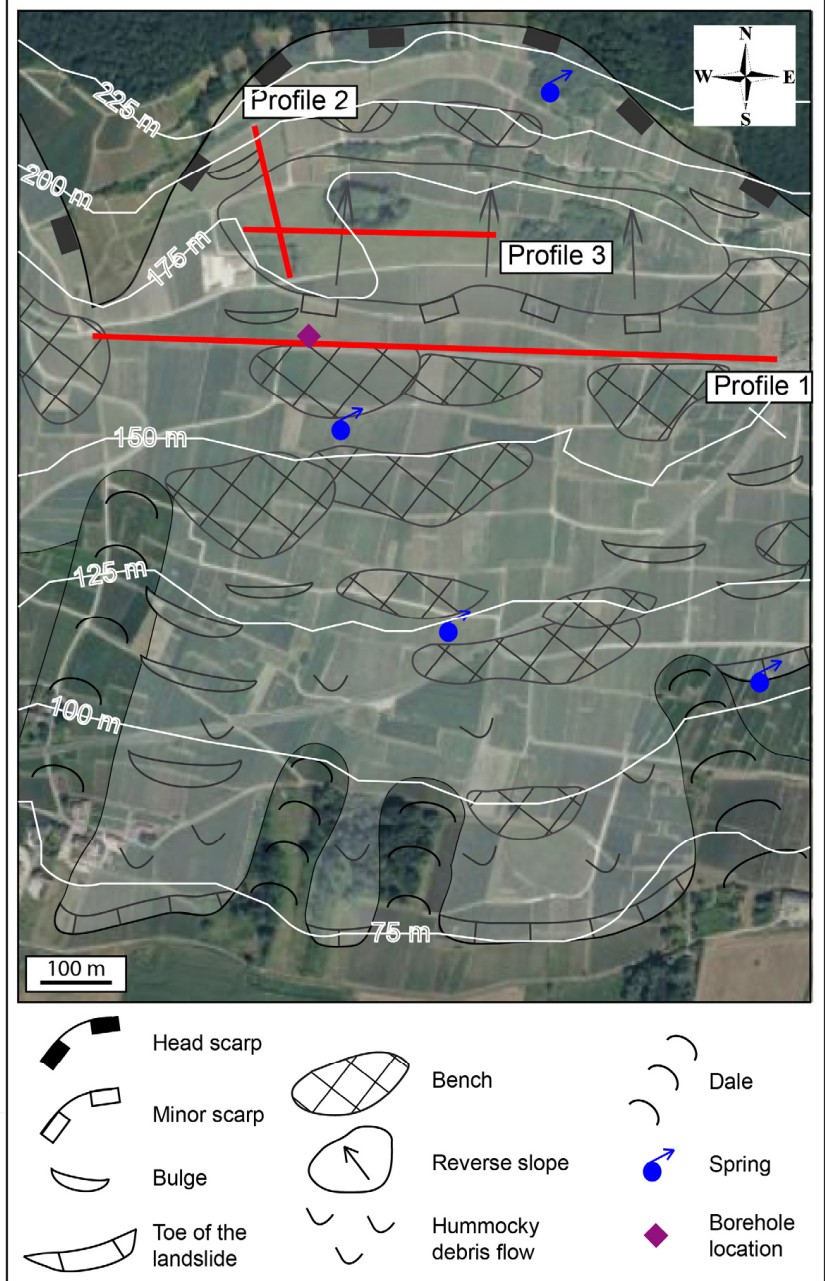

**Figure 3.** Geomorphological map of the Jacotines landslide and geophysical profiles locations.

To invert traveltimes we used the approach proposed by Gance et al. [37] where Fresnel volumes are used to regularize wavepath in the model. A representation of the wavepath coverage (see Results section) shows the places where wavepaths's coverage is high (in red) or low (in blue). In the final tomograms (Figure 4), the place with no wavepaths are blanked and will ignored during the inversion and fusion processes. The size of these volumes depends on the main frequency of the seismic signal—taken here as 40 Hz—so that the quasi-Newton inversion scheme is stabilized and converge toward the global minimum of the cost function. To optimize the convergence process, we maximized the likelihood function defined as:

$$L = exp\left(\frac{-\sum_N \frac{tc-to}{\sigma}}{2}\right)$$

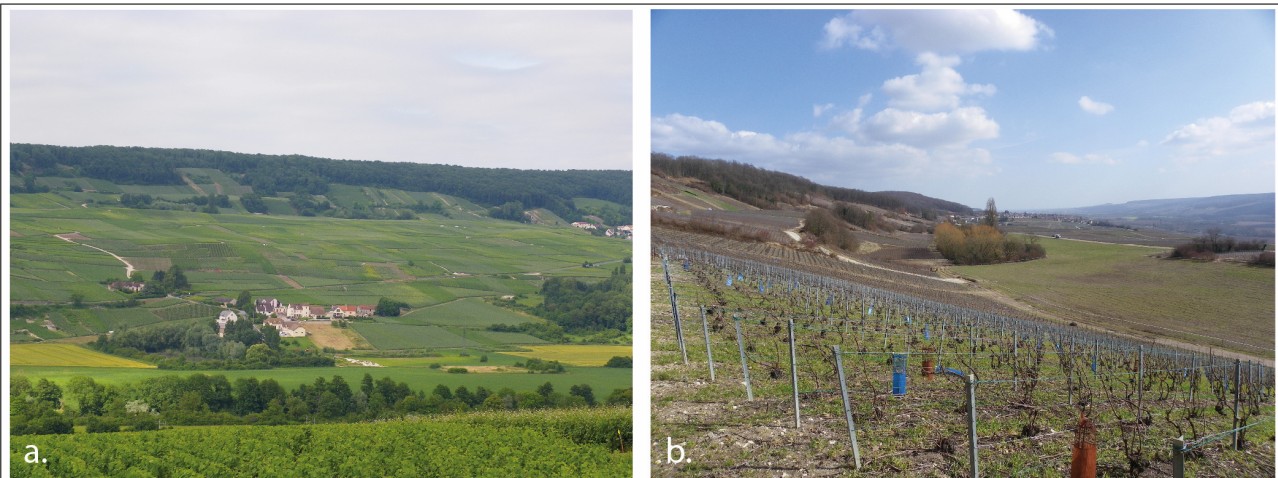

**Figure 4.** The Jacotines landslide from the opposite valley side (**a**) and side view of the head scarp (**b**).

*N* is the number of considered wavepaths, *tc* and *to* are calculate and observed propagating time respectively, s is the time picking a priori uncertainty—taken here as 0.01 s. In order to avoid overfitting, as well as underfitting, we decided to stop the iterative process when *L* reach 0.9. To invert apparent resistivities, we used the Res2dinv software (version 3.59) [38] based on a least squares/quasi-Newton inversion approach also able to damp the convergence process in order to avoid local minimum solutions.

Data fusion based on fuzzy subsets theory was realized to interpret jointly seismic (Vp) and electrical (ρ) tomograms in regard to physical behaviors and physical properties of the materials [39]. In the fusion process, the definition of belonging functions is determinant because they link the values of the geophysical parameters (Vp, rho) to the possibility that a specific geotechnical behavior occurs. Those functions where estimated from field observations where geologists can observe the consequences of this behavior (i.e., fissures, cracks) at the surface and estimate the corresponding inverted velocity or resistivity on the tomograms. This fusion approach allows to combine different possibility tomograms, but also the a posteriori uncertainty data issued from the inversion algorithms, increasing the reliability of the resulting images [23].

The results were complemented by a 60 m deep borehole along profile 1. The borehole location (Figure 3) was determined using preliminary results from tomography while also taking into account technical feasibility linked to the presence of the vineyard. This borehole crosses all the formations investigated by the geophysical surveys down to the cretaceous chalk.

## 3. Results

The geomorphological analysis gives an updated description of the slope conditions (Figure 3). The Jacotines landslide extends from the top of the hillslope (245 m a.s.l.) to the Marne valley floor (70 m a.s.l.). The main scarp forms a vast 75 m high amphitheater-like re-entrant at the edge of the plateau (Figure 4a). Its slope is partially graded (mainly by periglacial slope deposits) but may exceed 11°. At the foot of the main scarp, a first step stretches over 720 m (width), 210 m (depth), at an average height of 170 m a.s.l. It is the only place without vineyard because of the bogs located at the foot of the slip face, a typical feature of rotational slip. Downslope, a partial dislocation of this slab determines the presence of two secondary steps, the first one at an elevation of about 160 m a.s.l. and the second one at 145 m a.s.l. Further down, a bulge covered with hydrophilic vegetation shows superficial reactivation. The toe of the landslide is 5 m high above the valley floor (Figure 4b). The Jacotines landslide is delimited by dales and is coalescent with adjacent landslides; it also reshapes an older landslide (probably of ante-Weichselian age) presenting a graded main scarp.

Geophysical profiles were exploited separately to understand the internal structure of the slope. We assume here that Vp parameter provides information on the variation of fissure density and on the presence of weathered material whereas the ρ parameter provides information on the variation of water content within and next to the landslide [39]. Along profile 2 (longitudinal, Figure 5), which extends between the foot of the main scarp and the external edge of the reverse slope, the Vp are greater than 2100 m/s at a depth of 40 or 50 m (i.e., 150 m a.s.l.). A few meters below the topographic surface, Vp range between 1100 and 2000 m/s. In the segment between points 100 and 160 m (horizontal distance), Vp range between 2000 and 2300 m/s. Near the topographic surface, velocities are below 900 m/s. Profile 3, transverse, is located on the main reverse slope. Below the elevation of 140 m a.s.l., Vp are greater than 2100 m/s, like on the profile 1. Between elevations of 140 and 160 m a.s.l., velocities are more contrasted (ranging between 1100 and 2300 m/s). Near the topographic surface, Vp are less than 900 m/s (Figure 5). Along profile 1, Vp are measured to a depth of up to 100 m. However, this depth is not reached in some places due to a background noise caused by nearby works in the vineyard (e.g., profile 1, Figure 5). The Vp are the highest below 110 m where speed values exceed 2700 m/s. They range between 1700 and 2500 m/s (110–140 m a.s.l.) and 1100 and 1700 m/s (elevation up to 160 m a.s.l.). Near the surface, they are lower than 1000 m/s.

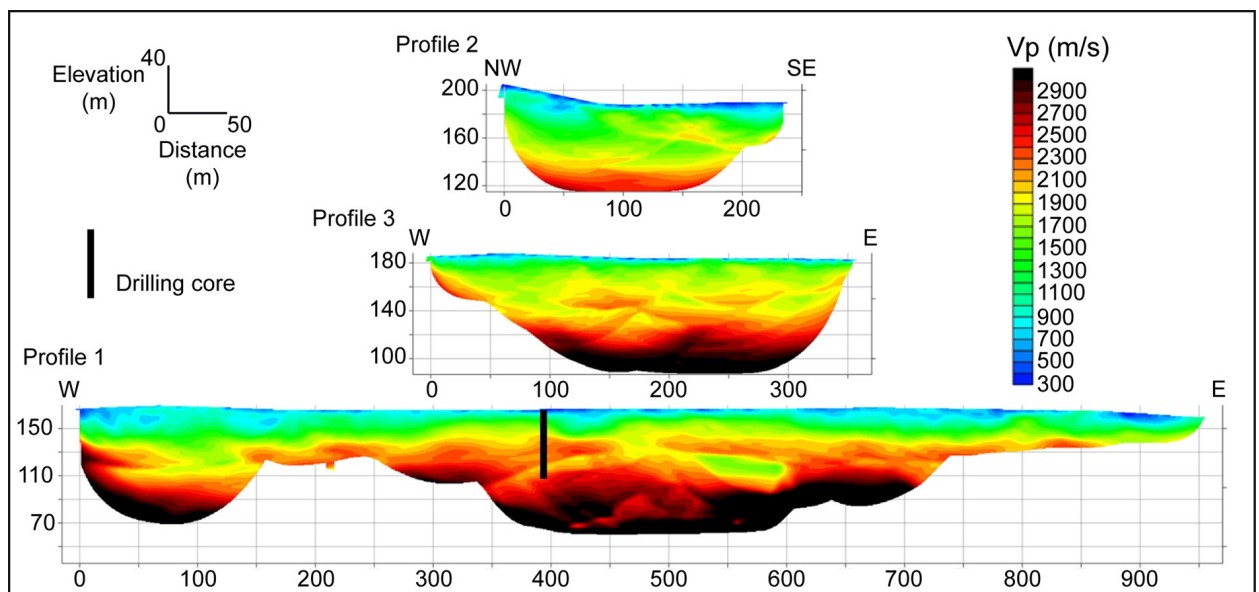

**Figure 5.** Seismic wave velocity, profiles 1, 2 and 3.

Consequently, at about 110 m a.s.l. a very high Vp level corresponds to the presence of a chalky layer (Cretaceous). This compact level is overlain by a more heterogeneous material up to about 10 m below the surface that can be divided into two parts, the highest velocities being encountered in the deepest part. Finally, near surface low propagation velocities could indicate sheared and fissured materials as well as colluvial materials.

The ρ tomogram shows various tendencies (profile 1, Figure 6). The first ten meters below the surface are featured by resistivity values lower than 30 Ohm.m. This characteristic is almost continuous along the profile. Between 10 and 20 m deep, the ρ is regularly greater than 80 Ohm.m. However, more conductive volumes may be inserted at this level. Between 20 and 50 m deep, i.e., at elevations comprised between 110 and 140 m a.s.l., the ρ is lower and does not exceed 20 Ohm.m. Finally, deeper than 50 m, below an elevation of 110 m a.s.l., the resistivity is greater than 100 Ohm.m. Thus, the ρ tomogram reveals limits similar to those appearing on the Vp tomogram.

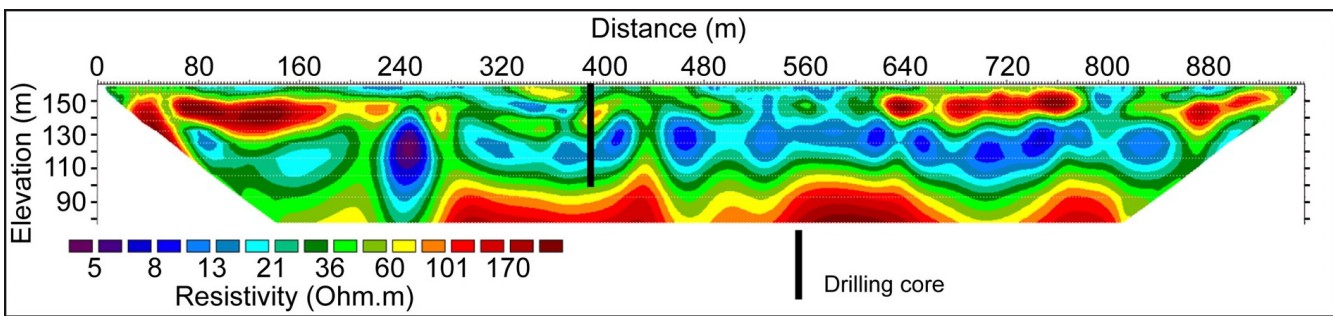

**Figure 6.** Electrical resistivity (ρ), profile 1.

Data fusion was finally used to verify and interpret the similarities between seismic and resistivity data as a result of water-saturation and shear characteristics of the material. Given the inherent uncertainty of the method [39] the fusion integrates the likelihood distribution in order to check the reliability of each kind of data, i.e., Vp and ρ (Figure 7a,b). The likelihood functions show that the seismic data are not sufficiently accurate in the first 20 m and beyond a depth of 60 m (Figure 7a), where the density of seismic ray is insufficient. Conversely, electrical data are most reliable near the surface (Figure 7b), as shown by the computation of a sensitivity matrix. Thus, the combination of seismic and resistivity tomograms gives reliable results up to a depth of 60 m. The conversion from geophysical parameters to ground properties is based on the determination of belonging functions (Figure 7c,d). This determination was done by comparing the inverted resistivity and velocity values at the surface with what we observed on the field in terms of rock saturation and fissuring. In the Jacotines landslide case, wave velocities lower than 700 m/s indicate on the field disturbed material due to shearing or surface weathering, while values greater than 1500 m/s indicate sound material (Figure 7c) [40]. Concerning resistivity, when the ρ values are less than 50 Ohm.m, the material is observed as saturated, whereas above 200 Ohm.m it appears as rather dry (Figure 7d). Below and beyond these limits the rock can be considered as fissured/saturated and sound and dry respectively. Between them, we can extrapolate linearly intermediate behaviors, more or less disturbed and saturated. These functions are then used to convert tomogram values in terms of possibility—between 0 and 1—to get a fissured or saturated material representations. As described by Grandjean et al. [39], the fusion is then operated by using fuzzy operator theory in which we use the sup() function to take the maximal value of the two possibility sections (Figure 7a,b up) after we modulate them by the likelihood (Figure 7a,b down) related to the inversion uncertainties. Finally, data fusion (Figure 7e) allows to check whether the hypothesis of a water-saturated ground or of a sheared material is true (= 1) or false (= 0).

Data fusion highlights the consistency of the ρ and Vp results. It presents a sheared level with high quantity of water near the surface and a rather heterogeneous material between 10 to 50 m depth (i.e., at an elevation of 110 and 150 m a.s.l.). The lower part appears less sheared and/or saturated than the upper part (the transition being around 123 m a.s.l.; Figure 7e). Homogeneous levels appear at a depth of investigation greater than 50 m. They are moderately sheared and/or saturated.

To confirm these results, a borehole was drilled along profile 1, where the reliability of the geophysical investigations had been established. Technically, the drilling location requires an unplanted and free of artifact (e.g., drainage system) vineyard plot, like the one located around the meter point 400 (horizontal distance on profile 1; Figure 5). The purpose is to attribute a facies to the different geophysical responses (Figure 8).

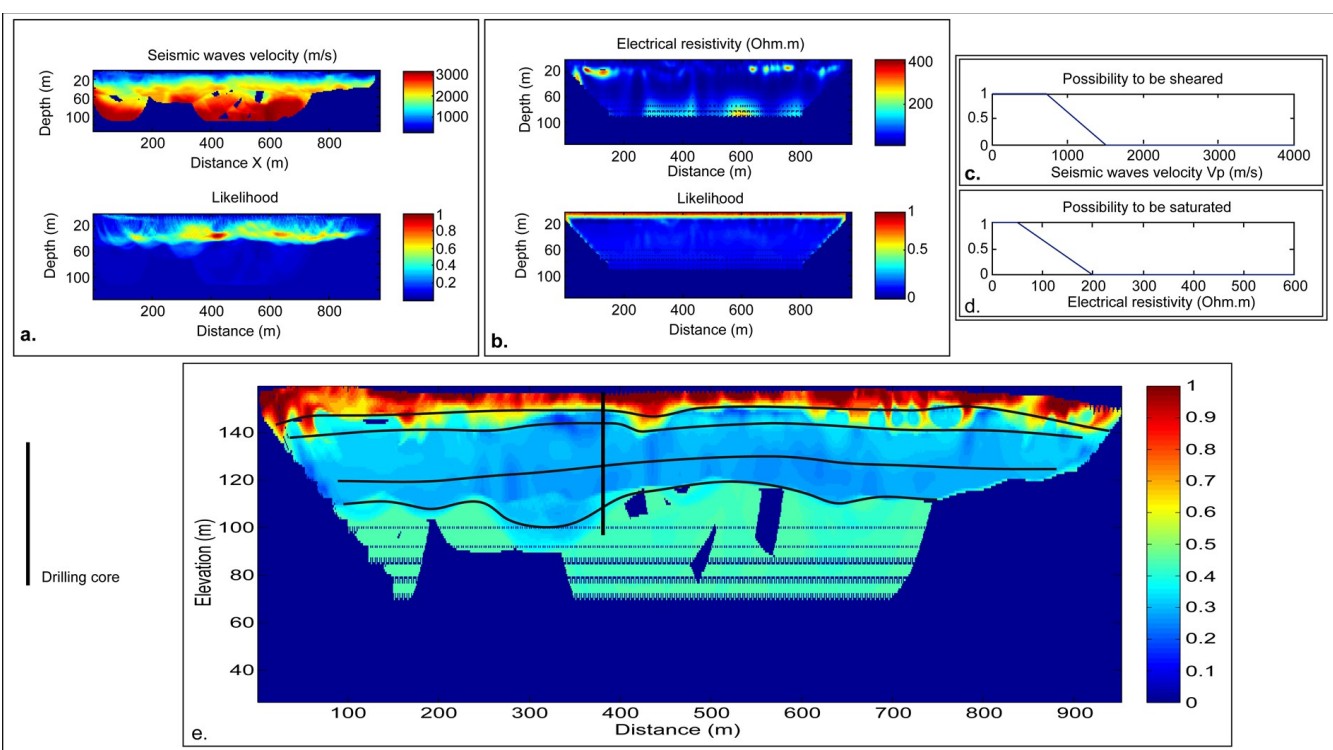

**Figure 7.** Profile 1: Likelihood distribution of seismic data (**a**) and electrical data (**b**); belonging functions (**c**,**d**); fuzzy logic fusion tomogram with main discontinuities (**e**).

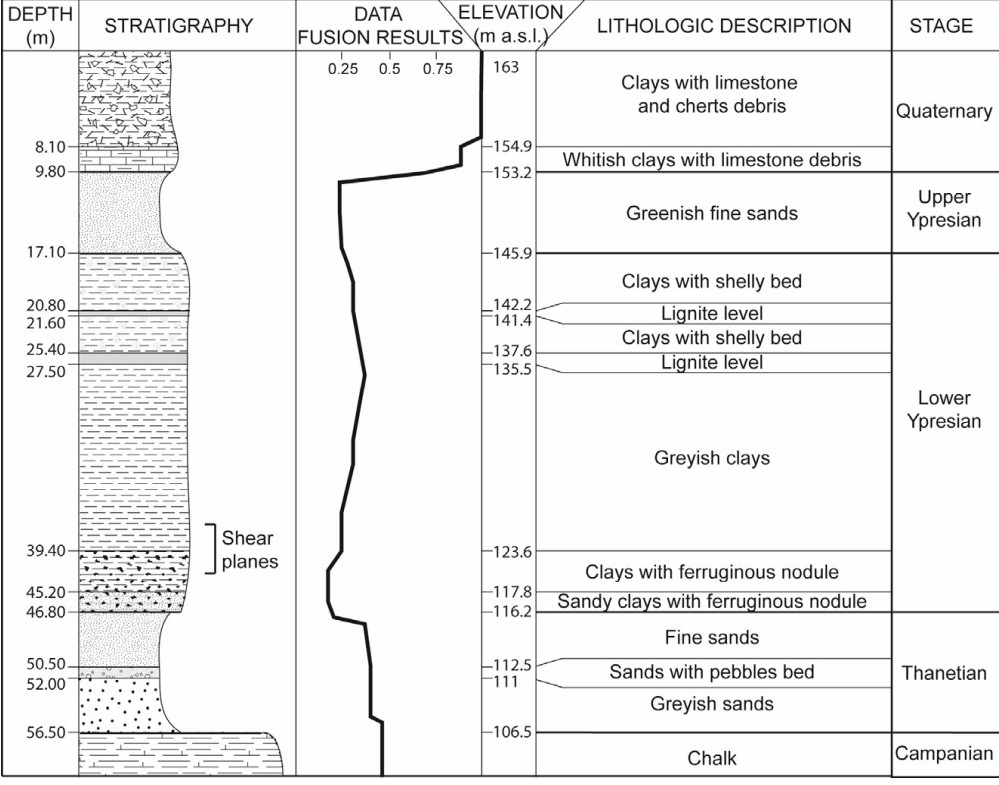

**Figure 8.** Correlation between lithostratigraphy (drilling core) and data fusion result.

From top to bottom, the lithostratigraphy is as follow: (1) 10 m of heterogeneous displaced material; (2) 7 m thick layer of fine sand; (3) 10 m of shelly clays with intercalation of lignite beds. These clays can be sandy, especially between 21 and 24 m deep; (4) 12 m of

green and then greyish clays; (5) 7 m of clays and clayey sands with ferruginous nodules, and with carbonation along shear planes; (6) 10 m of sand, with an intercalation of flint pebbles of centimeter scale; (7) in situ white chalk.

The comparison of the drilling core lithostratigraphy with the data of the geological map and other bibliographic references for this area shows that all the cenozoïc strata are found in their normal succession, although the boundary between the Thanetian and the Ypresian formations and the one between the Lower Ypresian and the Upper Ypresian formations are difficult to determine here. Moreover, the Lower Ypresian formation is 28 m thick in the borehole, while its maximum thickness is normally 25 m [41]. Conversely, the thickness of the Upper Ypresian, 7 m, seems a little weak. The five shear planes that follow one another between a depth of 37.5 m and 42 m might explain these thickness anomalies (Figure 8). The combination of the drilling core description and the geophysical data makes it possible to define the different elements of the data fusion tomogram. The discontinuities identified through geophysics (Figure 7e) are clearly associated with facies changes (Figure 8) and can be extrapolated to the entire profile.

## 4. Discussion

An interpretative profile of the landslide can be drawn from the synthesis of all the data collected (Figure 9). Thus, the evolution of the Jacotines landslide can be reconstituted.

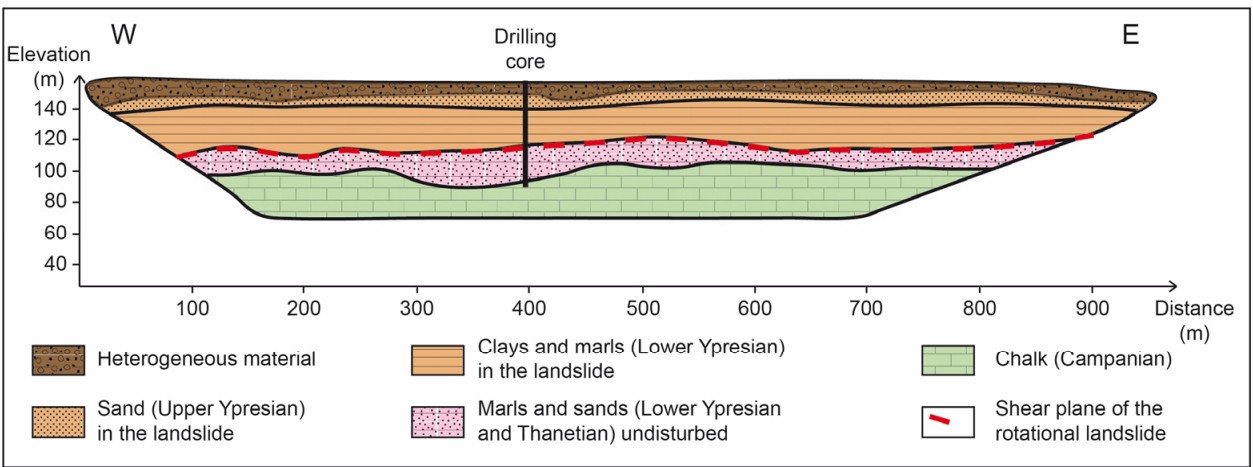

**Figure 9.** Interpretive transverse profile on the basis of the geophysical data (profile 1).

Surface landform analysis, borehole data and geophysical investigations allow to determine the thickness of the landslide. The geological layers located below a depth of 40 m, between elevations of 110 and 120 m a.s.l., are undisturbed. From bottom to top, the Cretaceous chalk is overlain by the sandy formations of the Thanetian and of the Lower Ypresian. The five shear planes observed in Lower Ypresian clays at about a 40 m depth correspond to the sliding surface of the Jacotines landslide. Over these shear planes all the layers are displaced and correspond to the landslide affected strata. Within the displaced body the Ypresian formations retain their original stratigraphy. On the contrary, the heterogeneous material present in the first 10 upper meters consists of a disorganized blend of strata (clays of the Lower Ypresian, sand of the Upper Ypresian, "Marnes et caillasses" of the Lutetian; Figure 10) which outcrop higher on the hillslope. It results from a recent debris flow superimposed on the rotational landslide.

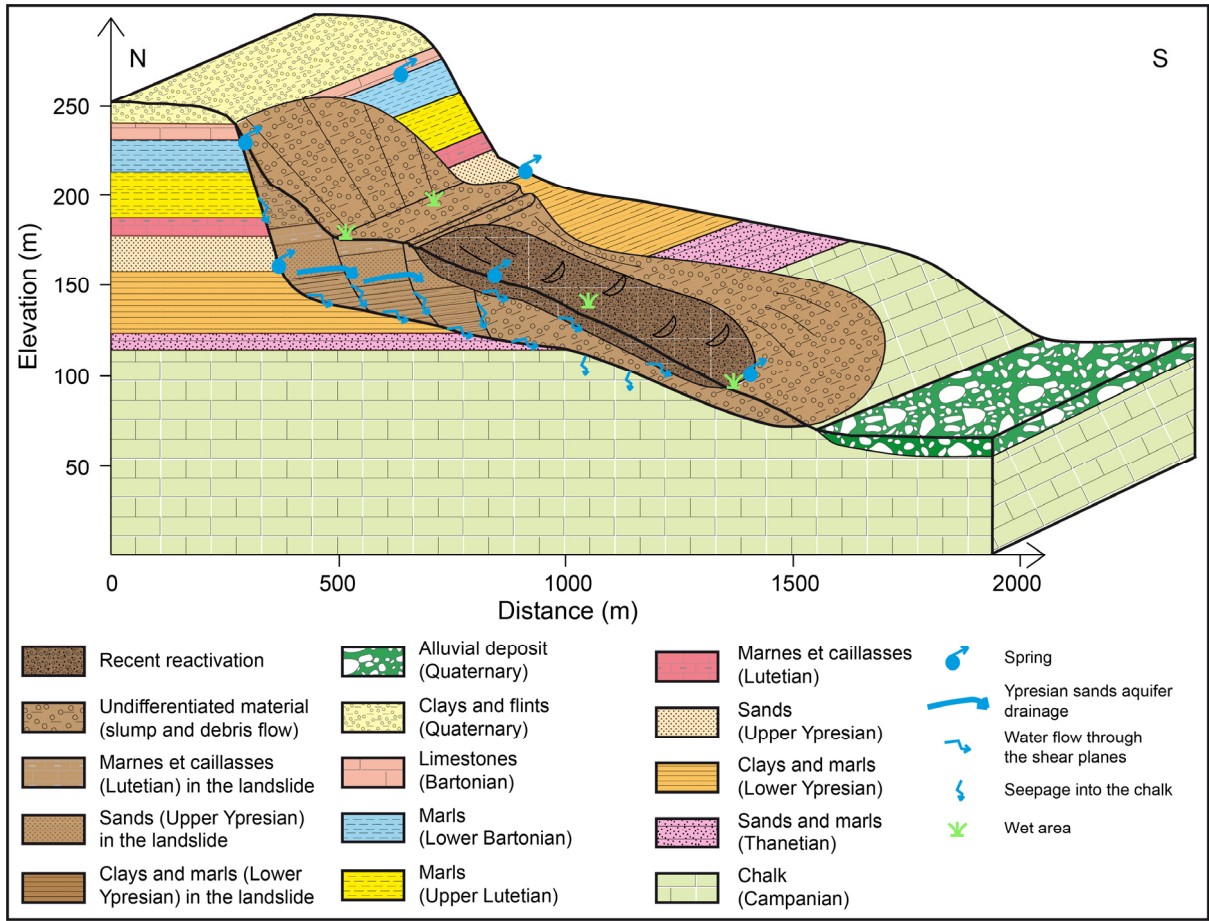

**Figure 10.** Water flow pattern in the Jacotines landslide.

The cross-correlation of the data allows reconstructing a model of the Jacotines landslide (Figure 10). According to the analysis of the surface topography, this debris flow is 700 m long. Along this section, the instability evidences are numerous (bulges, solifluxion, sinuous alignment of vineyards …) and reveal recent or, if they are slow, current movements. In the upper part of the landslide, the shear plane position used in the model is defined by the interpretation of the seismic profiles 2 and 3. The surface steps quoted on the geomorphological map (Figure 3) correspond at depth to the dislocation in three distinct parts of the sliding mass. A little lower down the slope, a fourth step is likely, but remains to be confirmed (no geophysical measurements were performed at that location).

The drilling core confirms the two distinct generations of movement (in addition to older instabilities—probably dating back to the last glacial maximum or even earlier—indicated by graded head scarps in the upper part of the hillslope) (Figure 10). Considering the water required for such a rotational landslide, it may be attributed to Lateglacial or Subboreal periods with excessive water balance. Recent activity near the surface consists of a remobilization of sliding materials. Superficial reactivations are still active as illustrated by geomorphological and hydrologic features (bulge, water retained in hollows) and by geophysical data (structural degradation and water content of the materials). These reactivations are the most common cause of damage in the Champagne vineyards. Triggering them is a complex process that cannot be explained directly by heavy rainfall events [42] but rather by the water content of the sliding masses which is controlled by the slow draining of the Ypresian aquifer. Thus, mass movements do not contain a transmissive aquifer with rheocrene springs: the heterogeneity of the disturbed material impedes the water circulation [10,16,39].

Along the Marne valley, more than 30% of the total surface of hillslopes between Dormans and Epernay are affected by Lateglacial landslides, 32.7% on the slopes facing north and 30.9% on the slopes facing south (Figure 1). In the same area, only 12.8% of the springs (inventory of the data base of the French geological survey, BRGM) are located within the slide masses, and 63.4% at their direct periphery (head scarps and toe of the landslides). This distribution (Table 1) reflects the interrelation between mass movements and the draining of the aquifer.

**Table 1.** Spatial distribution of springs in the Marne valley according to their geomorphological context.

| Geomorphological Context | Number of Springs | % |
|---|---|---|
| head scarp | 38 | 40.4 |
| slide mass | 12 | 12.8 |
| toe of landslide | 22 | 23.4 |
| rocky slope | 22 | 23.4 |
| total | 94 | 100 |

At the top of the hillslope the Upper Ypresian sand and the Bartonian limestones aquifers are exposed, which explains the numerous springs at this location (Figures 3 and 10). The spring located at the foot of the reverse slopes of the upper part of the landslide (elevation 155 m a.s.l., Figure 10) is supplied by the semi-confined aquifers of the upper units of the slump in which the geological structure is retained. Conversely, the few springs in the debris flow reveal the relative impermeability of this unit. Nevertheless, increased interstitial pressure along discontinuities (Figure 10) promotes superficial reactivation [10]. Springs at the toe of the landslide are fed by slow drainage along the discontinuities between ancient and recent debris flow units. Moreover, in the case of the Jacotines landslide, most of the water flowing along the surface that separates the debris flows and the bedrock is drained towards the cretaceous chalk aquifer [10]. This explains the small amount of springs at the toe of the landslide, which is therefore less exposed to reactivation. As a matter of fact, among the thirty-five active landslides identified in the area (data base of the BRGM), only 14% exhibit reactivations on the lower part of the hillslope. Both mechanical and hydrogeological characteristics of the chalk substratum clearly enhanced the stability of overlying slope deposits.

## 5. Conclusions

The internal structure of the Jacotines landslide is revealed by an approach combining geomorphology, geophysics and geology. Each method confirms or specifies the assumptions derived from the other two. They particularly allow understanding the interactions between sliding bodies (present on more than 30% of the hillslopes of the area) and groundwater supply, which is essential to the knowledge of the superficial reactivation mechanisms leading to instability [43]. Landslides are located where groundwater supply is abundant [10] and the spatial distribution of the springs is conditioned by the presence of large landslides that act as semi-permeable cover, implying a shift of the spring line downslope. The drainage of the Eocene aquifer maintains a permanent, although variable moisture content in slope deposits that promotes reactivation. However, where the slope deposits overlap the Campanian chalk, they are partially drained downward, and therefore more stable.

The current landforms are the result of two main phases of instability. The first one generated a rotational slip in morphoclimatic context characterized by an excess of water, probably during late glacial time. The second one leads to superficial reactivations which may be still active and related to present conditions. The case study of the Jacotines landslide is especially interesting since it is a representative example in the Champagne vineyards. Thus, the conclusions drawn from the study of its internal structure and its hydrogeological behavior can be extrapolated to similar landslides in the area or to

landslides in similar geological structures. This methodology is an essential step to manage the risk of mass movement, which is a central objective in the Champagne vineyards considering the economic stakes. This work, supplemented by geotechnical study and piezometric monitoring [16], is being used as a basis for the study of sliding susceptibility of the hillslopes of the vineyards. To better understand the dynamics of the Jacotines landslide and in particular its re-activation, high-resolution landform monitoring (DTM) over a long period of time is required. Taken as a pilot site, the landslide is now being monitored by Lidar, which shows continuous readjustments over short time intervals (work in progress). It should make it possible to determine the volume of the slipped masses in correlation with the human forcing and the climatic data. The installation of piezometers on the landslide should also allow to evaluate the influence of the water content and circulation paths on the landslide activity. These results should convince winegrowers of the benefits of the "stabilization by hydro-geological action" [11] within and below the slipped mass rather than traditional surface drainage (e.g., draining trenches) which often only shifts the problem to adjacent plots. Finally, this device would help winegrowers to modify their agricultural practices for a better resilience to climate change.

**Author Contributions:** Conceptualization, N.B. and G.P.; methodology, N.B., G.G., G.F. and A.D.; validation, N.B., G.P. and G.G.; formal analysis, G.G.; investigation, N.B., G.F. and O.L.; data curation, N.B., G.P., G.G. and G.F.; writing—original draft preparation, N.B. and G.P.; writing—review and editing, N.B., G.P. and G.G.; supervision, A.D. All authors have read and agreed to the published version of the manuscript.

**Funding:** This research received no external funding.

**Data Availability Statement:** The data presented in this study are available on request from the corresponding author.

**Acknowledgments:** This work was done in collaboration with the Bureau de Recherches Géologiques et Minières (BRGM) and the Comité interprofessionnel du vin de Champagne (CIVC). The authors would like to thank A. Marre for fruitful discussions and K. Samyn for geophysical field measurements.

**Conflicts of Interest:** The authors declare no conflict of interest.

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
