# Peer review of "Internal Structure and Reactivations of a Mass Movement: The Case Study of the Jacotines Landslide (Champagne Vineyards, France)"

_2624-795X, doi:10.3390/geohazards4020011_

Round 1

Reviewer 1 Report

In the present paper, the authors reconstruct the relations between the landslide structure, surface water flow, groundwater regime and the overall slope stability based on the integration of geophysical imaging (seismic wave velocity and electrical resistivity), lithostratigraphic analysis (drilling core) and geomorphological investigations (surface landforms). A first phase of instability resulting in a large rotational slip probably occurred during the Late Glacial Period in morphoclimatic conditions characterized by an excess of water. A second one, still active, leading to superficial reactivations and relates to present hydrogeological conditions determined by the internal structure of the landslide. In general, it is well organized and written. It could be accepted for publication after the following concerns are fixed.

Detailed comments:

1)     (profile 2, 250 long; Figure 2) should be “(profile 2, 250 m long; Figure 2)”.

2)     “A representation of the wave-path coverage is shown in Figure 5a”. Only Figs. 1 and 2 are appeared before this statement. The Figures should be sequentially numbered at the appearance of Figures.

3)     Line 95. Do not indent “where” after the equations.

4)     The “40.4%, 12.8%, 23.4%, 100%” should be “40.4, 12.8, 23.4, 100” in Table 1 since “%” is specified in the table cell.

5)     “The first meters below the surface are featured by resistivity values lower than 30 Ohm.m”. Should it be “The first 10 meters …”?

6)     Change “(Figure 5c and 5d)” to “(Figures 5c and 5d)”.

7)   How did the authors obtain Fig. 5, especially Figs 5c and 5d? The authors should describe it in detail. How was data fusion performed? It should be described in detail.

Author Response

Please find below our answers to your remarks:

  1. Done.
  2. Done.
  3. Done.
  4. Done.
  5. Done.
  6. Done.
  7. Clarification have been made: lines 192 to 207.

Thank you for your comments.

Nicolas Bollot

Reviewer 2 Report

Please consider  comments in the attached file

Author Response

Please find below our answers to your remarks:

Introduction

We have specified the objectives and added 2 bibliographic references.

Study area and methods

We added the location of the study area and a new figure with geological map.

Results

We have inserted contourline in the map and a new figure with 2 pictures of the landslide.

Discussions

Our discussion concerns precisely the Jacotines landslide. We now refer to 2 references (16 and 42), one of which is new.

References

We have added 2 new regional references (16, 17). They bring a contextualization in the specific problematic of our article. They also show all the interest of a multi approaches methodology such as the one developed here.

Thank you for your comments.

Nicolas Bollot

Reviewer 3 Report

This study proposed an integrated method to reconstruct the relations between the landslide structure, surface water flow, and groundwater regime. The Jacotines landslide is also an interesting area for the landslide study. However, this study has provided insufficient discussion on the novelty and effectiveness of the proposed methodology. Major revisions should be made prior to final acceptance. Here listed are my comments:

1. This research lacks a comprehensive review of past research methods in the introduction section, as well as a comparison between the research methods proposed in this study and past methods. Is this difference innovative?

For example, the authors mentioned “The approach presented is based on the integration of geophysical imaging (seismic wave velocity and electrical resistivity), lithostratigraphic analysis (drilling core) and geomorphological investigations (surface landforms) to reconstruct the relations between the landslide structure, surface water flow, groundwater regime and the overall slope stability.”. Is there a new technique in these works?

2. This study used geophysical imaging, lithostratigraphic analysis, and geomorphological investigations to estimate the geological structures. Generally, in order to prove the accuracy of the estimation, one or two verification wells are drilled for proof. Is there any direct evidence in this study to demonstrate that your estimation is correct?

3. How did this study directly map the results from Figure 7 to Figure 8, and what methods and evidence were used? The discussion in the text is only briefly mentioned in lines 240-248 without any further explanation. The profile 2 proposed in this study is only near the scarp, so how was it possible to know the subsurface conditions of the toe? Additionally, Figure 8 shows that all the strata have a horizontal dip angle. Is this an assumption made in this study, or is there direct evidence for this?

4. This article reads more like a geological survey project report, simply outlining the work carried out and the potential estimation results. However, as a research article, readers would like to know if the authors have proposed any new survey techniques, and if so, how these new techniques differ from previous ones and whether they improve the accuracy of survey results.

Author Response

Please find below our answers to your remarks:

  1. In the introduction, it is specified that the (known) techniques used have never been associated (in Champagne), and indeed are rarely associated elsewhere. This is now mentioned in the text.
  2. The reviewer suggests that we produce “one or two boreholes” to check our interpretation: in fact our interpretation is based, among other things, on a borehole (figure 8, see also figures 3, 5, 6, 7 and 9).
  3. Figures 7 and 8 (now: 9 and 10) are based on the interpretation of surface morphology, geophysical data and borehole data (line 250), but also on the geological map (Hatrival, 1977; it is an aclinal sedimentary structure, see line 64 and Figure 8). In Figure 10, the thickness of the debris flow is a very likely approximation, and given the lithostratigraphy (Figure 8) it can only lie on the substratum (in this case, chalk). On this issue, a new reference has been added: Bollot et al., 2022.
  4. The results presented are therefore not estimates, they are based on a crossing of methods never applied in the Champagne vineyard and are original and unpublished.

Round 2

Reviewer 2 Report

I have reviewed the manuscript “Internal structure and reactivations of a mass movement: the case study of the Jacotines landslide (Champagne vineyards, France)”. Based on the changes made, the work is complete. I believe that the work is a good contribution to the landslide analisys integrating different methodology. The work is of high potential with very good data and well presented.

I recommend improving the geological map. Add contourline and arrange the legend (make it neater)

I can only congratulate the authors for the manuscript.

Reviewer 3 Report

The paper has been improved, which could be published in its current form.